# BENTO: BENCHMARKING CLASSICAL AND AI DOCKING ON DRUG DESIGN–RELEVANT DATA

## ABSTRACT

Recent advances in artificial intelligence have introduced deep learning and co-folding approaches for predicting protein-ligand complexes, raising the question of their applicability and how they compare with classical docking methods. In this work, we present a thorough benchmarking study of eleven tools for protein-ligand interaction prediction, spanning classical molecular docking methods, deep learning-based models, and co-folding algorithms. While most related benchmarking efforts primarily assess the generalization capacity, we extend the analysis to also evaluate the performance on drug design-relevant data and across different classes of protein-ligand complexes. Here, we introduce BENTO, a comprehensive benchmark that evaluates 11 tools for protein-ligand interaction prediction – both established and recently developed – across four test datasets and multiple derived subsets in a pocket-aware setup. We show that 1) careful dataset curation is essential – filtering by pocket structural similarity and controlling ligand complexity exposes generalization failures that are obscured in conventional benchmarks; 2) classical and deep learning-based docking tools perform similarly well on drug-like ligands, making them comparably useful for virtual screening, with physics-based methods offering a clear advantage in speed; 3) co-folding tools outperform other approaches on structurally complex ligands, whereas most methods achieve similar accuracy on regular small molecules; and 4) all methods struggle to generalize to unseen pockets, with deep learning models being the most prone to overfitting. Overall, our results show that while current docking and DL-based approaches are reliable for many drug-design-relevant scenarios, genuine pocket-level generalization remains an open challenge. BENTO provides a rigorous and transparent framework for diagnosing these limitations and guiding the development of more robust protein-ligand prediction models.

## 1 INTRODUCTION

The field of protein-ligand interaction prediction is rapidly evolving, with new tools emerging at a steady pace. This progress highlights the need for comprehensive and unbiased benchmarking to evaluate the preformance of tools, explore their limitations, and assess their applicability.

Recent benchmarking efforts have provided important insights into both classical and deep learning (DL) docking methods, guiding further tool development and evaluation of their applicability. For example, PoseBusters (Buttenschoen et al., 2024) showed that DL-based docking approaches frequently fail to generate chemically valid structures and often underperform compared to physics-based tools when both validity and root mean square deviation (RMSD) are considered. Physical validity, however, can often be restored through post-processing with energy minimization.

Most benchmarks primarily assess tool generalization to unseen proteins, often by introducing new datasets. DockGen (Corso et al., 2024) demonstrated that relying solely on protein sequence similarity for train-test splitting is insufficient and emphasized the importance of considering binding-pocket similarity. To address this, DockGen introduced a validation dataset enriched with complexes containing binding pockets distinct from those in PDBBind (Liu et al., 2017) and Binding MOAD (Wagle et al., 2023), a commonly used training resource for DL docking.

The most recent benchmark, PoseBench(Morehead et al., 2024), compared co-folding methods with both classical and DL docking approaches. Results showed that co-folding generally outperforms

conventional and DL-based docking baselines but still struggles to predict complexes involving novel protein-ligand binding poses, with AlphaFold3 achieving the best overall performance.

Here, we introduce BENTO, a benchmark that systematically evaluates 11 diverse tools for protein-ligand complex prediction, spanning classical methods, purely DL-based docking, and co-folding approaches, including several recent methods not covered in prior benchmarks (Section 2). Unlike earlier efforts, BENTO not only uses established test sets but also introduces curated subsets stratified by ligand structural and physicochemical properties, as well as by protein and ligand similarity, to better understand how these factors influence the tool performance. Importantly, we evaluate ligand- and protein-related effects separately to avoid confounding.

Our results reveal that commonly used test sets vary widely in ligand structural and physicochemical properties, as well as in protein and ligand similarity to the training sets. These differences explain much of the variation in tool performance across datasets. We find that AlphaFold3 generally outperforms other methods on complex ligands with higher structural complexity, while for typical small-molecule ligands, several classical, DL-based, and co-folding approaches perform comparably. For drug-like data involving proteins similar to common training sets, DL-based docking performs best, while AlphaFold3 and classical methods rank lower. However, when dissimilar binding pockets are considered, performance drops substantially across nearly all methods – except for Gnina. Most DL docking tools show limited generalization ability and appear strongly overfitted to training data. Co-folding methods are more robust but still face challenges.

Our main contributions:

- We demonstrate the importance of careful dataset curation and stratification along multiple criteria when working with multifactor data such as protein-ligand complexes. In particular, when constructing a dataset for evaluating generalization to unseen pockets, proteins should be filtered explicitly based on pocket structural similarity rather than by PDB release date. Moreover, ligand complexity should be controlled separately, as it can confound assessments of pocket-level generalization.

- Both classical and DL-based docking tools perform well on drug-design-relevant data, making them suitable for practical therapeutic development. However, under the pocket-aware setting, all tools perform equally poorly on protein pockets that are dissimilar to those in their training data.

- For structurally complex ligands, co-folding tools generally outperform other approaches, whereas most tools achieve comparable accuracy on regular small-molecule ligands.

## 2 RELATED WORK

**PoseBusters**  PoseBusters (Buttenschoen et al., 2024) evaluates seven docking methods—Gold (Verdonk et al., 2003), Vina (Trott & Olson, 2009), DeepDock (Méndez-Lucio et al., 2021), Uni-Mol (Zhou et al., 2023), DiffDock (Corso et al., 2023), EquiBind (Stärk et al., 2022), and TankBind (Lu et al., 2022)—on the Astex dataset (Hartshorn et al., 2007) and the newly introduced PoseBusters dataset. The benchmark shows that (i) classical docking tools continue to outperform contemporary deep learning approaches in terms of physical plausibility and generalization to unseen cases, and (ii) post-prediction minimization can markedly improve the quality of DL-generated poses. A limitation, however, is that the "unseen" split is defined solely by PDB release date, without accounting for protein-pocket or ligand similarity to the training data. Subsequent benchmarks have evaluated more advanced DL methods, including co-folding approaches, which have been shown to surpass purely physics-based methods.

**DockGen**  DockGen (Corso et al., 2024) examines seven methods – Vina, smina (Koes et al., 2013), Gnina (McNutt et al., 2025), two versions of DiffDock (Corso et al. (2023), Corso et al. (2024)), EquiBind, and TankBind – using the DockGen dataset. The authors find that DL models fail to generalize to unseen binding modes primarily due to overfitting to the training distribution. However, DockGen's ligand set is relatively narrow, which may confound conclusions about pocket-level generalization, and the dataset also contains known annotation errors. Moreover, the benchmark defines the search space without using the true ligand as a reference, potentially underestimating achievable performance.

**PoseBench** PoseBench (Morehead et al., 2024) assesses eight methods – Vina, DiffDock, DynamicBind (Lu et al., 2024), NeuralPLexer (Qiao et al., 2024a), RosettaFold All-Atom (Krishna et al., 2024), Chai1 (Boitreaud et al., 2024), Boltz-1 (Wohlwend et al., 2024), and AlphaFold3 (Abramson et al., 2024) – across PoseBusters, Astex, DockGen, and CASP15 (Robin et al., 2023). The benchmark reports that 1) co-folding approaches generally outperform traditional and standalone DL docking methods, 2) even state-of-the-art tools struggle to predict binding modes for genuinely unseen targets, 3) some co-folding models are highly sensitive to multiple sequence alignment quality, while others show greater robustness, and 4) current DL methods still struggle to balance structural accuracy with chemical specificity. Limitations include the fact that generalization is assessed using datasets not explicitly curated to ensure protein dissimilarity, and that, similar to DockGen, the search space definition does not use the true ligand as a pocket reference.

**PoseX** The recent PoseX benchmark (Jiang et al., 2025) evaluates 23 docking tools and introduces a new dataset designed for both self-docking and cross-docking. A major strength of the study is its careful curation of pocket dissimilarity: pocket structures in the new dataset are explicitly compared, via structural alignment, to all protein-ligand complexes released prior to 2022, ensuring a more meaningful assessment of generalization. The authors report two notable findings. Firstly, deep learning and co-folding methods outperform physics-based approaches in overall docking success rates in cross-docking, though the observation is expected given that previous works reported superiority of DL methods over physics-based in self-docking scenarios. Secondly, nearly all co-folding methods exhibit systematic issues with ligand chirality.

Our benchmark, Bento, evaluates 11 diverse docking tools, making it the second-largest benchmark after PoseX. Although PoseX is the largest in scale, it includes several methods that have been repeatedly shown to perform poorly in independent studies; therefore, we excluded those tools from our evaluation. In contrast to prior benchmarks, Bento not only incorporates all widely used test datasets (PoseBusters, Astex, DockGen, and PDBBind Timesplit) but also stratifies them into subsets based on ligand properties as well as ligand and protein similarity to training data. This allows us to examine method performance across realistic application scenarios – for example, on drug-like ligands. To our knowledge, Bento is the first benchmark to analyze ligand- and protein-related factors independently, ensuring that their effects on generalization are not conflated.

## 3 MATERIALS AND METHODS

### 3.1 DOCKING TOOLS AND THEIR LAUNCH

We evaluated 11 widely used tools for predicting protein-ligand interactions, encompassing both classical physics-based docking methods and state-of-the-art neural network models. The classical category includes AutoDock Vina and smina. Neural network-based approaches can be broadly divided into two groups: (a) models that take a protein structure as input to predict ligand binding poses, and (b) co-folding methods that directly generate protein-ligand complex structures from a protein sequence and ligand structure. The first group includes the physics-augmented convolutional neural network (CNN) Gnina, the latest version of DiffDock, Uni-Mol Docking V2 (Alcaide et al., 2024), NeuralPLexer, and two recently introduced flow-matching-based models, FlowDock (Morehead & Cheng, 2025) and Matcha (Anonymous, ICLR 2026, under review). An anonymized version of the manuscript describing Matcha is provided in the supplementary material. The co-folding category comprises AlphaFold3, Boltz-2, and Chai-1. Table 1 summarizes the algorithms underlying each tool, their ability to operate in blind-docking and pocket-restricted modes, and their input requirements. Tools that accept a binding site reference were run using the geometric center of mass of the true ligand: Vina, smina, Gnina, Uni-Mol Docking V2, and Matcha. See more details in Appendix B.4.

### 3.2 TEST DATASETS

We used established datasets for evaluation of protein-ligand interaction prediction tools: Timesplit test (Corso et al., 2023) of PDBBind (Liu et al., 2017), PoseBusters (Buttenschoen et al., 2024) (version 2), Astex (Hartshorn et al., 2007), and DockGen (Corso et al., 2024).

Table 1: Docking tools and their modes benchmarked in the study. 3D – three-dimensional, GNN – graph neural network. All: Ligand / Pocket / Blind.

| Tool | Under the hood | Modes | Protein input |
|---|---|---|---|
| **Physics-based tools** | | | |
| AutoDock Vina | Empirical scoring | All | 3D structure |
| smina | Empirical scoring | All | 3D structure |
| **DL tools** | | | |
| gnina | Vina sampling + 3D CNN rescoring | All | 3D structure |
| DiffDock | GNN + Riemannian diffusion | Blind | 3D structure |
| Uni-Mol Docking V2 | SE(3)-equivariant transformer | Ligand | 3D structure |
| NeuralPLexer | Transformer (no diffusion) | Blind | 3D structure |
| FlowDock | Generative model based on conditional flow matching | Blind | Sequence |
| Matcha | Transformer (DiT-inspired) + Riemannian flow matching | Ligand / Blind | 3D structure |
| **DL tools (co-folding)** | | | |
| AlphaFold3 | Euclidean diffusion | Blind | Sequences |
| Boltz-2 | Euclidean diffusion | Pocket | Sequences |
| Chai-1 | Euclidean diffusion | Blind | Sequences |

The DockGen dataset was designed to include proteins with binding pockets dissimilar to those in PDBBind train set and Binding MOAD (Wagle et al., 2023), further referred as training sets. However, its approach relies on protein domain clustering, which is not a suitable proxy for binding-pocket classification: domain clustering reflects global fold similarity rather than the local structural environment that determines ligand binding. Another limitation concerns the Timesplit test set, which has been reported to suffer from data leakage (Li et al., 2023). Nevertheless, we included both datasets in our study, as we combined all available test sets and then re-partitioned them using our leakage-free procedure. This procedure ensures splitting based on both binding-pocket and ligand similarity to the training sets (see Section 3.3). The datasets sizes and processing procedures are summarized in Appendix A, Table 2.

### 3.3 SUBSETS

From all combined test sets we extract distinct subsets combined by criteria (Figure 1): a) Subsets by ligand class; b) Subsets by ligand complexity (size, flexibility and burial in the binding pocket); c) Drug-like subsets; d) Subsets by protein pockets similarity; e) Subsets by ligands similarity. Inclusion criteria are described in Appendix B.1.

## 4 RESULTS AND DISCUSSION

### 4.1 BENCHMARK SETUP

We evaluated the performance of eleven tools for predicting protein-ligand complex structures, including both widely used physics-based molecular docking methods (Vina, smina) and diverse deep learning-based approaches: Gnina, DiffDock, Uni-Mol Docking V2, NeuralPLexer, and the most recent FlowDock and Matcha, as well as co-folding methods such as AlphaFold3, Boltz-2, Chai-1. For benchmarking, we used established datasets: PDBBind Timesplit test, PoseBusters, Astex, and DockGen. After filtering, we compiled a combined set of 1,047 complexes, further referred as the Tests, and also constructed focused subsets based on specific criteria (Methods 3.2,3.3).

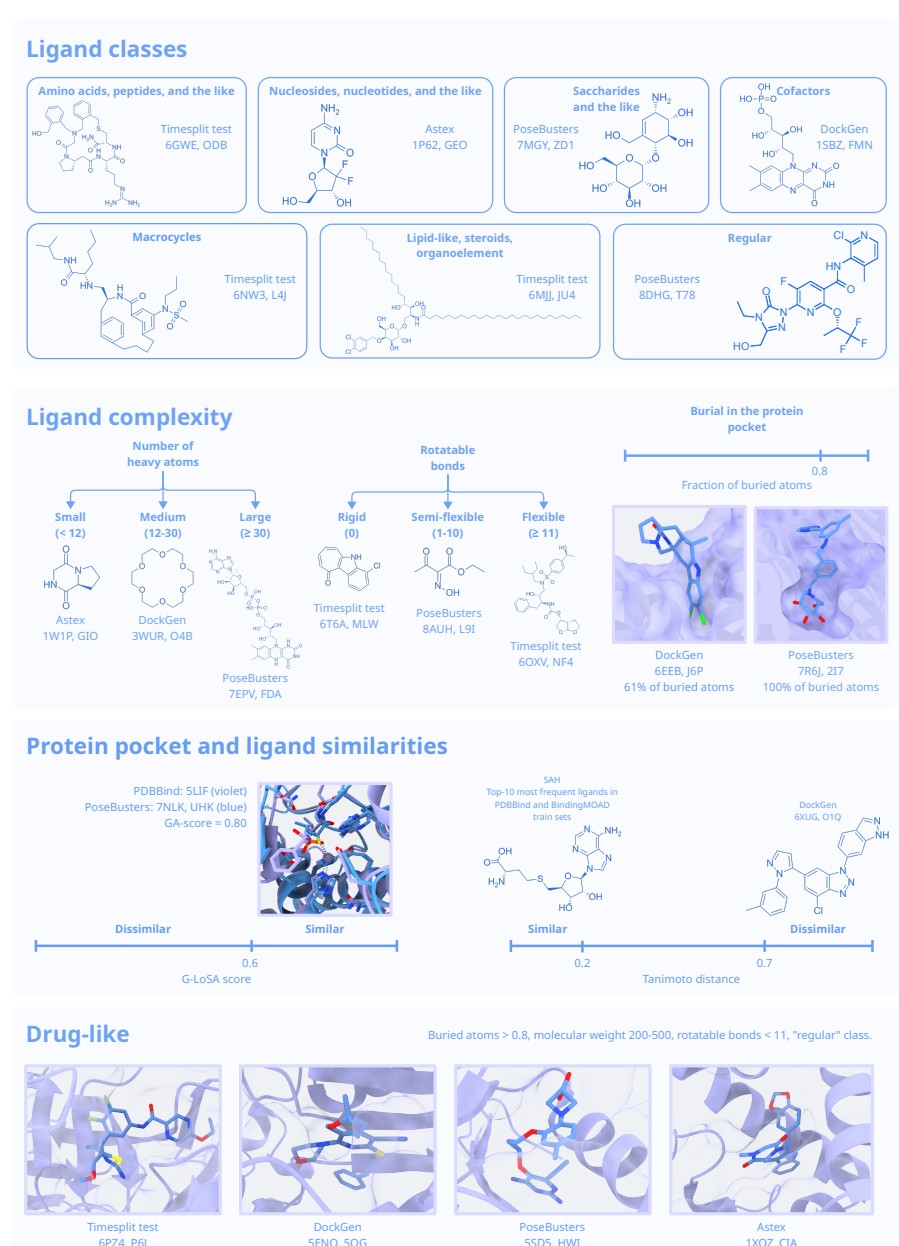

Figure 1: Data subsets used in the study.

Our baseline is designed to comprehensively evaluate protein-ligand interaction prediction tools and to address the following key questions:

- What is the generalization ability of the tools? How well do these methods perform on ligands and proteins that are distinct from the training sets commonly used?

- How do the tools perform on targets and ligands typical of drug discovery? While benchmarks often emphasize performance on unseen data, it is equally important to assess their practical utility in real-world drug development tasks.

- How do the tools handle different classes of compounds and ligands with diverse physico-chemical properties?

To address the first two questions, we extracted subsets of complexes with varying levels of ligand and pocket similarity to common training sets, quantified using Tanimoto similarity and G-LoSA alignment score (GA-score), respectively (Appendix B.2, B.3).

To address the third question, we annotated ligands in the test sets according to common chemical classes: (1) amino acids, peptides, and peptide-like molecules; (2) nucleosides, nucleotides, and the like; (3) saccharides and the like; (4) cofactors; (5) macrocycles; and (6) lipid-like molecules, steroids, and organoelement compounds (grouped together due to their scarcity). Ligands not belonging to these categories were considered "regular" small molecules. The detailed annotation procedure is described in Appendix B.1.

For evaluation, we used the percentage of predictions with RMSD ≤ 2Å from all predictions for the given set. We also assess physical-validity using PoseBusters validity criterion (PB-valid). See details in Appendix B.4.

Differences in tool performance were evaluated using pairwise two-sample tests for equality of proportions. Tools were then grouped such that all members within a group showed no statistically significant differences in success rate. The grouped bars in Figures 2, 4-6, and 8-9 reflect these statistically indistinguishable sets of tools.

## 4.2 Tools performance on diverse types of data

**Performance on test datasets: tools rank differently on individual datasets** First, we evaluated tool performance on the filtered benchmark datasets. Consistent with previous studies, the highest performance was observed on the Astex dataset, with the recent DL-tool Matcha emerging as the top performer alongside with Gnina. In contrast, the most challenging dataset proved to be DockGen, which was specifically designed to include dissimilar proteins. Here, the co-folding methods AlphaFold3 and Chai-1 ranked highest, although their performance remained modest, with only 14% valid predictions. The weakest results were obtained from DL-tools Uni-Mol Docking V2, NeuralPLexer, and FlowDock and, therefore, these methods were excluded from subsequent analyses. On the PoseBusters dataset – considered the most stringent quality benchmark – Gnina achieved the best performance, followed by classical smina, and co-folding tools – AlphaFold3 and Boltz-2. As highlighted in the original PoseBusters study, physics-based methods and the physics-augmented Gnina generally produce more physically valid predictions than purely deep learning-based approaches.

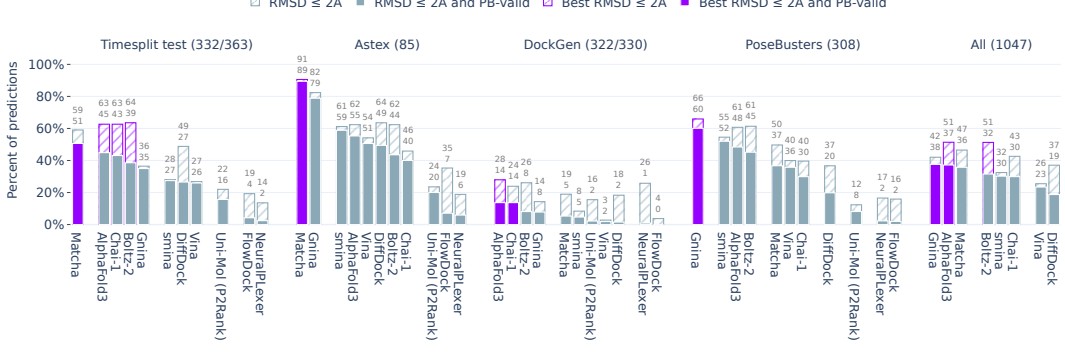

Figure 2: Tools performance on test sets.

We next conducted a more detailed analysis of the datasets to assess whether factors beyond protein dissimilarity contribute to performance differences.

**Datasets vary in ligand types, ligand physicochemical properties, and similarity to training sets** As shown in Figure 3A, the ligand composition varies substantially across datasets. Timesplit test and PoseBusters primarily contain regular ligands, with moderate representation of other classes. Astex is the most homogeneous dataset, dominated by regular small-molecule ligands typical of drug candidate design. By contrast, DockGen is the most diverse, with comparable representation

of all ligand classes. This ligand composition helps explain the high tool performance on Astex and the relatively poor performance on DockGen.

In terms of ligand size and flexibility (Figure 3B, middle and right panels), Astex consists mainly of small molecules of sizes typical for drug-like compounds. The other datasets also include larger, bulkier, and more flexible ligands, which are more challenging for prediction methods.

With respect to similarity to training sets (Figure 3C, left panel), PoseBusters, followed by DockGen, contain proteins with binding pockets most dissimilar to those in the training sets. In contrast, Astex and, to a lesser extent, Timesplit test, include proteins whose pockets are more similar to training examples. Interestingly, although DockGen was designed to emphasize proteins dissimilar to training sets, it contains the largest fraction of ligands similar to those in training, followed by Astex. Timesplit test and PoseBusters contain ligands most distinct from training data.

Overall, PoseBusters is the most distinct from training dataset, both in terms of ligands and protein pockets, while still being dominated by regular ligands with moderate representation of other classes. DockGen, in contrast, features highly diverse ligand classes but those that are more similar to training ligands, despite having distinct protein pockets. The strong performance observed on Astex can be attributed to its abundance of regular small molecules and high similarity to the training set.

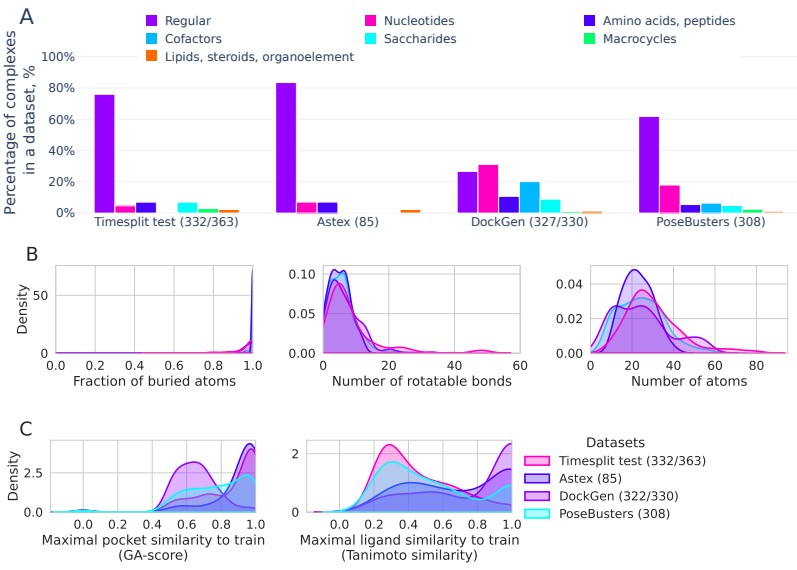

Figure 3: Overview of benchmark datasets. (A) Distribution of ligand classes across datasets. (B) Distributions of ligand physicochemical properties. (C) Distributions of maximal ligand and protein similarities to training sets. Similarities were computed in an all-against-all manner between the benchmark datasets and training sets; for each complex, the maximum similarity to the training set was taken.

**Performance on different ligand types: co-folding excels on complex compounds yet remains challenged** The datasets are heterogeneous with respect to the structural and physicochemical features of their ligands, prompting us to examine how tools perform across different ligand types.

As seen from Figure 4, for ligand classes typically characterized by greater structural complexity – such as peptide-like molecules, saccharide-like compounds, cofactors, and macrocycles – co-folding tools mostly outperform other methods. However, they still struggle to generate physically plausible predictions. By contrast, regular small-molecule ligands are easier to predict, with Gnina being the top method, followed by co-folding tools, Matcha, and smina.

It is important to note that ligands within the same class can vary considerably in size and complexity. For example, the "peptides" category includes both simple dipeptides and large macrocyclic peptides with non-standard units, while cofactors also show substantial structural diversity. To ac-

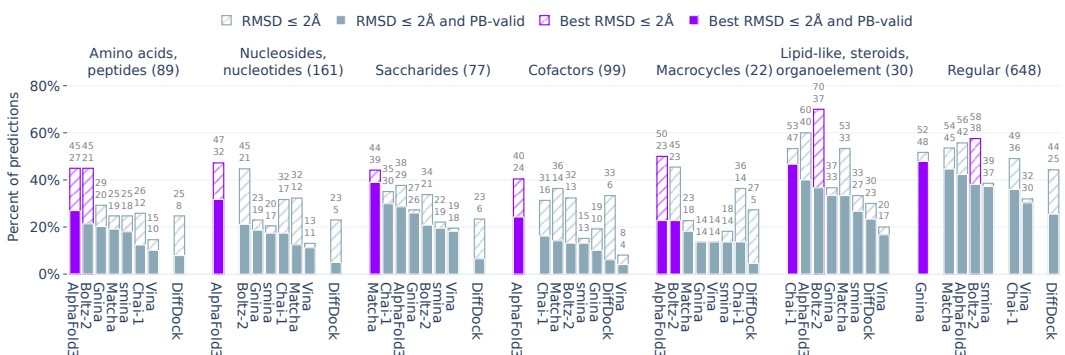

Figure 4: Tools performance of different ligand classes.

count for this, we further assessed tool performance on ligand subsets grouped by molecular size and conformational flexibility. As expected, ligands with high molecular weight and conformational freedom remain the most challenging, with AlphaFold3 and Gnina again performing best overall (Figure 8). Interestingly, when considering physically valid predictions, Gnina and AlphaFold3 often perform comparably – even though AlphaFold3 achieves a higher overall fraction of predictions below the RMSD threshold. This highlights a persistent limitation in the physical realism of predictions generated by current deep learning methods.

**Performance on drug-design material: Gnina ranks best** Most previous docking benchmarks have focused on evaluating performance on challenging, unseen data. However, it is equally important to assess how tools perform on data typical for drug discovery – namely ligands from screening libraries that may become drug candidates, as well as common protein targets such as kinases and frequent off-target proteins.

From the benchmark datasets, we compiled a subset representing drug design-relevant cases by restricting ligands based on number of atoms, conformational complexity, and burial within the protein pocket (Figure 5). On this subset, Gnina, Matcha, and AlphaFold3 perform equally if RMSD alone is considered, but physical validity slightly drops in the row.

We also extracted a drug design-like subset from PoseBusters – the highest-quality dataset considered – by applying drug-like criteria to this dataset (Figure 5). On this subset, tools generally produced a higher fraction of physically valid predictions, with Gnina reaching 70% PB-valid predictions. Notably, smina outperformed both co-folding and purely deep learning methods, highlighting the continued value of physics-aware docking approaches for drug-relevant tasks.

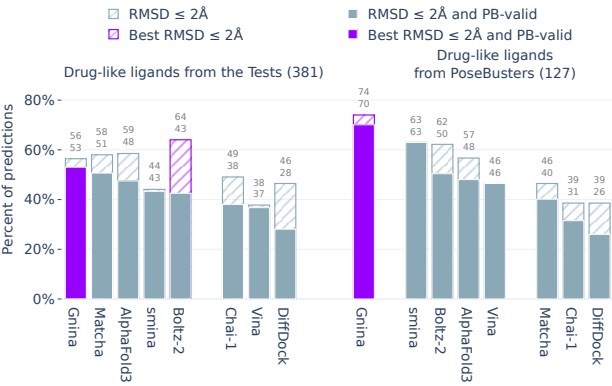

Figure 5: Tools performance on drug-like ligands from the Tests and PoseBusters.

**Performance on unseen proteins and ligands: limitations of co-folding and the need for curated data** The challenge in unbiased assessment of protein-ligand interaction prediction tools is that

two factors must always be considered: the ligand and the protein. When compiling datasets to evaluate a tool's ability to generalize to novel proteins, it is essential to ensure that performance is not confounded by ligand-related effects. For example, as noted above, the drop in performance observed for some benchmarks may be explained by an overrepresentation of specific ligand classes with high structural complexity, as in the case of DockGen.

To address this, we evaluated the generalization to unseen protein pockets using only the subset of typical ligands from the benchmark datasets. This subset was further split into complexes with similar and dissimilar pockets (see Methods, Appendix B.2). While performance showed only a slight increase on proteins with similar pockets, a substantial drop was observed on dissimilar ones for both co-folding methods and classical docking, with an even more pronounced decline for purely deep learning-based tools (Figure 6). Importantly, we show that co-folding and physics-based tools perform equally poorly on dissimilar pockets, which is consistent with PoseX (Jiang et al. (2025) paragraph Performance Stratified by Pocket Similarity). It indicates that earlier benchmarks may have overestimated the capabilities of DL methods due to less stringent or absent procedures for constructing genuinely out-of-distribution test sets.

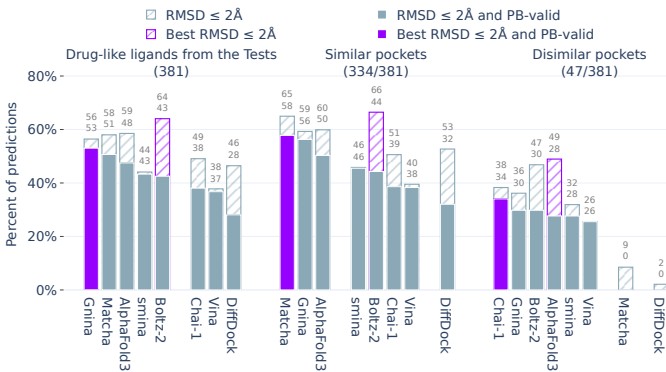

Figure 6: Tools performance on similar and dissimilar protein pockets from typical-ligands subset.

To assess the influence of ligand similarity on tool performance, we controlled for protein pocket similarity by selecting a subset of drug-like ligands bound to similar pockets (Figure 9). The performance gain on similar ligands is even more pronounced than in the case of similar pockets, with Gnina and Matcha again ranking highest. For dissimilar ligands, the performance drop is less severe than for dissimilar pockets, with most tools showing comparable results, except for DiffDock that is most affected.

## 5 CONCLUSION

In this work, we introduced BENTO, the most comprehensive benchmark to date for protein-ligand complex prediction, covering 11 diverse tools spanning classical docking, deep learning-based docking, and co-folding approaches, both widely used and recently released novel architectures. Unlike previous benchmarks, BENTO goes beyond standard test sets by introducing curated subsets stratified by ligand structural and physicochemical properties, as well as by protein and ligand similarity to training sets. This design allowed us to disentangle protein- and ligand-specific effects and to systematically study how these factors shape tool performance.

Our results reveal several notable trends. First, co-folding tools generally achieve the highest accuracy on structurally complex ligands, whereas small-molecule ligands are predicted with comparable quality across classical, DL-based, and co-folding methods. Second, DL-based docking tools such as Gnina and Matcha, the physics-based method smina, and co-folding approaches perform similarly well on drug-like ligands. This finding is of practical significance: these methods can be used for virtual screening with comparable expected accuracy, while physics-based tools retain a clear advantage in speed – an important factor in large-scale drug discovery workflows. Finally, all methods struggle to generalize to unseen pockets, with DL approaches being the most prone to overfitting.

Overall, Gnina demonstrates the highest robustness and accuracy and should therefore be considered essential in the benchmarking of novel tools. Among co-folding algorithms, AlphaFold3 is the strongest performer, although it is still outperformed by older methods in certain scenarios. Finally, BENTO highlights Matcha as a promising new DL-based approach.

To sum up, our study demonstrates that rigorous dataset curation is essential for benchmarking protein-ligand interaction prediction tools, given the multifactor nature of the task. We further show that overfitting remains a major limitation of current DL-based docking methods, restricting their generalizability. At the same time, typical drug-design data involving binding pockets similar to common training sets can be predicted reliably by several tools, underscoring their practical utility in such scenarios. Finally, our results highlight that incorporating physics-based principles into DL frameworks is crucial to enhance both generalization and predictive robustness.

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

# APPENDIX

## A    DATASETS

**Tests sets and their processing**    Table 2 summarizes used datasets and their sizes. We removed duplicated entries across datasets (PDB IDs: 1IG3, 6NP3, 6NPP, 6O70, 6XB3), complexes with peptides longer than eight amino acids, inorganic ligands (chemical component ID: 9UX), and entry that is not a protein-ligand complex (PDB ID: 2VDF from DockGen).

Table 2: Benchmark datasets used for evaluation.

| Dataset name | Size |
|---|---|
| Timesplit test | 332/363 |
| PoseBusters | 308 |
| Astex | 85 |
| DockGen | 322/330 |
| **Total** | **1047** |

**Training sets of assessed tools**    Matcha, FlowDock, and DiffDock were trained on both PDBBind and Binding MOAD. Uni-Mol Docking V2 was trained solely on Binding MOAD. Gnina relied on the CrossDocked2020 set (Francoeur et al., 2020), which was constructed from PDB structures and uses affinity annotations from PDBBind. NeuralPLexer was trained on its own curated dataset, PL2019–74k. The original paper also reports a NeuralPLexer variant trained on PDBBind, whose performance is comparable to the model trained on PL2019–74k (Qiao et al., 2024a).

# B    METHODS

## B.1    LIGANDS ANNOTATION

**Ligand classes**    All annotations by classes were performed using both PDB (Berman, 2000) annotations and SMILES Arbitrary Target Specification (SMARTS) patterns (Daylight Chemical Information Systems, n.d.). For peptides, an additional Python script based on RDKit (Landrum & Contributors, 2016) was employed to identify continuous chains of peptide bonds. In most cases, the SMARTS patterns represent the general molecular scaffold of each class, without specifying bond order or connectivity. For example, as shown in Figure 7 the lipid-like pattern defines a glycerol-like fragment (where oxygen atoms may be substituted by carbon, nitrogen, or sulfur) with at least two carbon tails (each of length $\geq 8$). The tails may be connected to the glycerol fragment via ester-like or ether-like linkages (i.e., the connector carbon atom may or may not have a double bond to oxygen, respectively). The carbon tails may contain any type of carbon-carbon bond (single, double, or triple).

Figure 7: Schematic representation of the "lipid-like" SMARTS pattern. Here, "*" denotes any molecular fragment, "," indicates logical OR, ";" indicates logical AND, "!" indicates logical NOT, "R" denotes ring membership, and "~" represents any bond type.

A complete list of subclasses and their corresponding SMARTS strings is provided in the Supplementary data.

**Ligand complexity**    As the structural complexity of a ligand increases, accurately predicting its binding pose becomes increasingly challenging. In particular, both the number of rotatable bonds and the total number of atoms are expected to strongly influence docking performance. To systematically investigate this effect, we divided the complexes in the Tests into groups according to ligand complexity:

- Drug-like: Ligands belonging to "regular" class, having at least 80% of atoms buried within the binding site, fewer than 11 rotatable bonds, and molecular weight 200-500.
- Big ligands: Ligands with $\geq 30$ heavy atoms.
- Medium ligands: Ligands with 12-30 heavy atoms.
- Small ligands: Ligands with $<12$ atoms.
- Flexible ligands: Ligands with $\geq 11$ rotatable bonds.
- Semi-flexible ligands: Ligands with 1–10 rotatable bonds.
- Rigid ligands: Ligands with no rotatable bonds.

## B.2    PROTEIN POCKETS SIMILARITY

A common strategy for ensuring unbiased evaluation and assessing the generalization ability of docking tools to novel proteins is to construct leakage-free datasets. This is typically achieved by filtering based on (a) the PDB (Berman, 2000) release date relative to the tool's release, as in the PDBBind Timesplit test and PoseBusters, or (b) low protein sequence or binding-pocket similarity to the training set, as in DockGen. However, such strategies substantially reduce both the size and the diversity of the evaluation datasets. Moreover, the release-date split does not guarantee that

newly released structures are biologically dissimilar to earlier ones. To address these limitations, we developed a generalized approach that can be applied to existing datasets to derive new, leakage-free splits without the need to construct entirely new datasets from scratch.

All-against-all binding pockets similarities between train and tests sets were quantified using G-LoSA software (Lee & Im, 2016). G-LoSA provides a sequence-order-independent alignment of local protein structures and outputs a GA-score, a normalized metric that evaluates similarity based on chemical features, independent of pocket size. The local binding site was defined to include all residues with at least one heavy atom within a 4.5 Å distance from any atom of the bound ligand. GA-score ranging from 0 (dissimilar) to 1 (identical) was used as a measure of similarity.

To define similar and dissimilar pocket subsets for each test complex, we used the maximum similarity to the training set. Complexes with a GA-score below 0.6 were labeled dissimilar, and those above this threshold were labeled similar.

### B.3 LIGANDS SIMILARITY

Tanimoto similarities were computed between all ligands from the PDBBind and Binding MOAD training datasets and tests ligands using the `datamol.similarity.cdist` function from the datamol (datamol-io, 2025) Python package with default settings. Molecular representations were generated as Morgan fingerprints via the `rdkit.Chem.rdFingerprintGenerator.GetMorganGenerator` method, with the following parameters: radius=3, fpSize=2048, includeChirality=False, useBondTypes=True, countSimulation=False, countBounds=None, atomInvariantsGenerator=None, bondInvariantsGenerator=None.

Ligand subsets were defined by maximum similarity to the training set: complexes with Tanimoto similarity $< 0.3$ were labeled dissimilar, while those with similarity $> 0.8$ were labeled similar.

### B.4 TOOLS LAUNCH AND METRICS CALCULATION

**Tools launch** For AlphaFold3, Boltz-2, and Chai-1, we selected the prediction with the highest confidence score among five model seeds, each generating five samples, using 10 recycling steps. When modeling receptor structures, all chains were retained, and only exact duplicate chains were removed. Multiple sequence alignments were computed with JackHMMER (Eddy, 2011).

NeuralPlexer, DiffDock, and Uni-Mol Docking V2 were launched using their default inference parameters. FlowDock used true holo protein structures as receptor templates. For all methods, the top-scoring sample was retained for analysis.

**Note about Uni-Mol Docking V2** All pocket-aware tools were run with the true ligand center as input, reflecting the real-world case where the binding site is known. Uni-Mol Docking V2, however, is not directly comparable in this setting. Unlike Vina, smina, Gnina, and Matcha, which use the pocket center only as a starting point, Uni-Mol Docking V2 cuts a fixed-radius pocket around the reference center. This rigid cropping leaks information about the true binding site and prevents exploration beyond the supplied location. To ensure fairness, we instead report Uni-Mol Docking V2 results using protein centers predicted by P2Rank (Krivák & Hoksza, 2018).

**Importance of pocket-alignment** For models that generate the structure of the entire complex, we adopt the symmetric RMSD protocol of Abramson et al. (2024). Because that work does not fully specify the alignment procedure, we outline below how we implement the base alignment:

1. We retain the primary protein chain containing the greatest number of atoms within 10 Å of the ligand.

2. The pocket is defined as all backbone $C\alpha$ atoms within 10 Å of any heavy atom of the reference ligand.

3. The reference pocket is then aligned to the full predicted protein structure using $C\alpha$ atoms in PyMOL with five refinement cycles (default).

An alternative, pocket-based alignment has been introduced by Qiao et al. (2024b). In this approach, the predicted pocket is defined relative to the predicted ligand, and each chain within this pocket is aligned to the reference. The chain with the lowest RMSD is selected. Unlike the base procedure, this method aligns pockets directly rather than the full protein, which helps reduce translational errors.

We advocate that the base alignment provides a more reliable evaluation. Pocket-based alignment can produce overly optimistic RMSD values, particularly for multi-chain proteins with several binding sites, where non-corresponding pockets may align by chance. This can obscure important docking failures, such as placing the ligand in the wrong binding site. These methodological differences explain the discrepancy between our metrics and those reported for co-folding methods in other benchmarks.

**Statistical comparison of tool success rates** Success rates were computed for each tool on a fixed evaluation subset. Pairwise differences between tools were assessed using a two-sample test for equality of proportions, and p-values were adjusted for multiple comparisons using the Holm method. Tools were considered statistically indistinguishable when their corrected p-values exceeded the significance threshold of 0.05. Based on these pairwise results, tools were grouped so that all members of a group showed no significant differences in success rate. The procedure also produced per-tool success rates and matrices of raw and corrected p-values.

## B.5 TOOLS PERFORMANCE

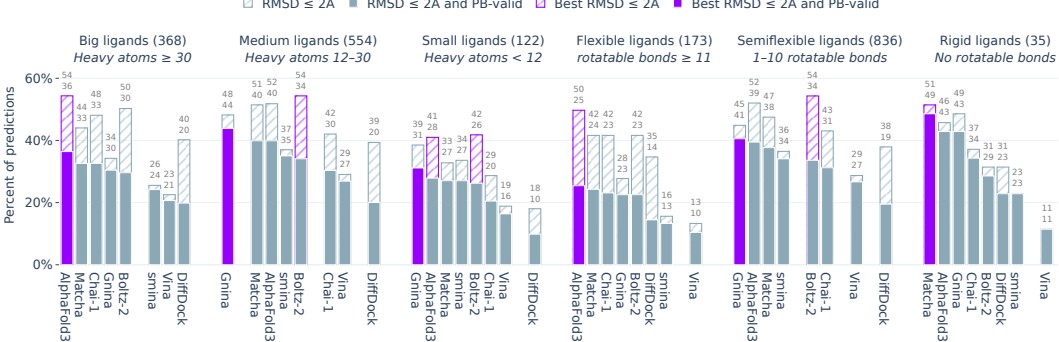

Figure 8: Tools performance on subsets grouped by ligands complexity.

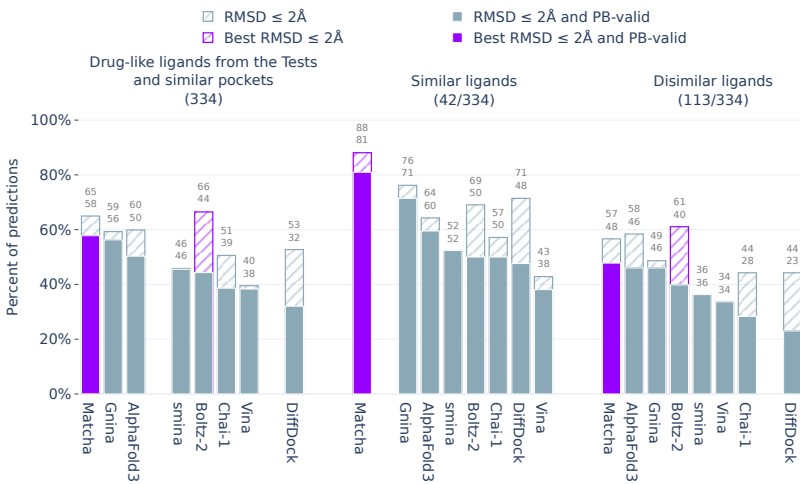

Figure 9: Tools performance on similar and dissimilar ligands from the subset of typical-ligands and similar pockets.

