# OpenReview forum: "Bento: Benchmarking Classical and AI Docking on Drug Design–Relevant Data"
_ICLR.cc/2026/Conference — Submitted to ICLR 2026_

### Official Review · Reviewer_xgwq · 2025-10-24

**Soundness:** 3
**Presentation:** 4
**Contribution:** 3
**Rating:** 6
**Confidence:** 5

**Summary:**

The authors introduce Bento, a new comprehensive benchmark for protein-ligand interaction prediction. The design and scope of this benchmark are compelling, and the authors' results and conclusions raise important questions for future work in developing deep learning models for protein-ligand interaction modeling.

**Strengths:**

1. The authors introduce the (now) largest benchmark for protein-ligand interaction prediction, which covers a wide variety of prediction use cases.
2. The authors' results and discussions highlight the importance of training new deep learning models with stronger generalization to novel protein binding pockets/interacting ligands.
3. The design of the benchmark's evaluation procedures follows best practices in the field.

**Weaknesses:**

1. As far as I can see, the authors haven't analyzed whether or how much deep learning methods' performance can be affected by the application of post-hoc structural relaxation via molecular dynamics software. This would be an interesting experiment to explore.
2. I did not find the benchmark's source code attached in the supplementary materials, which raises the question of whether the authors plan to open-source such code in the future. I would strongly encourage them to do so (if possible) and to prepare an online leaderboard as well.

**Questions:**

1. Is "This progress highlighting" a typo in Line 034?

---

> ### Author Response · Authors · 2025-12-02
>
> We thank the reviewer for valuable suggestions. Our point-by-point comments are below:
>
> **Q:** As far as I can see, the authors haven't analyzed whether or how much deep learning methods' performance can be affected by the application of post-hoc structural relaxation via molecular dynamics software. This would be an interesting experiment to explore.
> **A:** We thank the reviewer for highlighting this valuable point. We did not perform post-hoc structural relaxation analyses due to their substantial computational cost. Nevertheless, based on prior work, we expect that the percentage of valid predictions from DL tools would increase after post-prediction minimization. However, we do not anticipate this to substantially alter the main findings reported in the paper.
>
> ---
>
> **Q:** I did not find the benchmark's source code attached in the supplementary materials, which raises the question of whether the authors plan to open-source such code in the future. I would strongly encourage them to do so (if possible) and to prepare an online leaderboard as well.
> **A:** We provide the source code and have attached it to our submission.
>
> ---
>
> **Q:** Is "This progress highlighting" a typo in Line 034?
> **A:** Thank you for noticing. We have corrected the typo: "This progress highlighting" → "This progress highlights."

---

### Official Review · Reviewer_i6gM · 2025-10-30

**Soundness:** 2
**Presentation:** 2
**Contribution:** 2
**Rating:** 4
**Confidence:** 4

**Summary:**

The paper introduces BENTO, a benchmark of 11 protein–ligand pose prediction tools (classical docking, DL docking, and co-folding) across four public test sets, with curated subsets stratified by ligand class/complexity and by ligand/pocket similarity to "training sets" (PDBBind train and Binding MOAD). Authors found that AlphaFold3 performs best on structurally complex ligands. For typical small-molecule cases, several methods are comparable. While DL docking tends to overfit, Gnina appears most robust under pocket shift. By analyzing results of the models with various kinds of splitted benchmark datasets, the authors emphasize the need to evaluate not only on challenging OOD sets but also on homogeneous, drug-design–relevant subsets.

**Strengths:**

- **Well-scoped tool taxonomy and coverage.** The paper clearly separates DL docking (protein structure as input) from co-folding (protein sequence + ligand) and evaluates 11 tools spanning both categories.
- **Thoughtful dataset stratification.** Besides Timesplit/PoseBusters/Astex/DockGen, BENTO introduces subsets by ligand class, ligand complexity (size, rotatable bonds, burial), drug-like criteria, and pocket/ligand similarity using G-LoSA GA-score and Tanimoto thresholds.
- **Several interesting findings.** The highest ligand similarity in DockGen benchmark, good performance of Matcha (which predicts conformation based on translation, rotation, torsion Riemannian flow matching) in highly complex ligand molecules and etc.

**Weaknesses:**

- **Key experimental details are unspecified.** The paper does not elaborate on per-tool training dataset; "training sets" are treated at the benchmark level (PDBBind train/Binding MOAD/time-split criteria) for similarity analysis, making it hard to interpret generalization claims for each method. Also, details of running each tool are missing.
- **Pocket-conditioning realism and consistency.** Authors used P2Rank centers for Uni-Mol V2. However, when evaluating the performance of a pose prediction model, introducing P2Rank may create unintended biases unrelated to the model itself. Therefore, since models like Boltz-2 support pocket specification, it would be better to compare performance with and without pocket conditioning.
- **Lack of novelty.** Given that the paper synthesizes prior benchmarks rather than introducing a new one, the novelty is modest.

**Questions:**

1. **NeuralPLexer (NP2) categorization.** Table 1 places NeuralPLexer under DL docking rather than co-folding. As far as I know, NP2 can also generate the structure from sequence, so I've thought that this can be classified into co-folding model. Why did authors classified NP2 differently with the other co-folding models?
2. **Multi-ligand complexes.** How were cases with cofactors/ions/secondary ligands handled during inference and evaluation—were additional ligands kept, and if not, how might their absence affect PB-valid? (Only a single reference ligand and true-center usage are described.)
3. **Boltz-2 PB-validity.** PB-validity appears low for Boltz-2 in several subsets. This is directly opposite result compared to the Boltz-1x utilizing FK-steering with PB potential. Can the authors provide more details about running Boltz-2?
4. **Claim about Gnina’s training.** The text states Gnina "was not trained on PDBBind or Binding MOAD.", but they actually used their own CrossDocked dataset, which curated from PDBBind dataset. Please clarify the exact training dataset and provide a citation.
5. **Why Matcha excels on highly flexible ligands?** Can the authors provide an ablation/hypothesis (architecture, sampling, scoring) explaining Matcha's favorable behavior on flexible ligands? (Currently reported empirically without mechanism.)
6. **Details about experiment for different ligand classes.** How did authors predict the cofactor ligands? PoseBusters dataset contains many multi-ligand systems with cofactor and small molecule ligand. In this system, did authors only used cofactor, which consists of a single heavy atom, rather than using all non-polymer entities in the system?

## Typos
- Odd inequality symbols in the ligand-similarity thresholds in Appendix B.3.

---

> ### Author Response · Authors · 2025-12-02
>
> We would like to comment on the issues raised under *Weaknesses*.
> We have added information about the training sets used by the assessed tools in Appendix A, and details on running each tool are described in Appendix B.4. We thank the reviewer for noting that Boltz-2 supports pocket specification. We performed additional computations with pocket-conditioned Boltz-2 and updated our analyses accordingly. We have revised the Introduction and Conclusion to articulate our key findings more explicitly and added a Related Work section that outlines limitations of prior benchmarks and clarifies how our study differs.
>
> Below are detailed answers to the individual questions.
>
> **Q:** NeuralPLexer (NP2) categorization. Table 1 places NeuralPLexer under DL docking rather than co-folding. As far as I know, NP2 can also generate the structure from sequence, so I've thought that this can be classified into co-folding model. Why did authors classified NP2 differently with the other co-folding models?
>
> **A:** We used NeuralPLexer version 1 (http://dx.doi.org/10.1038/s42256-024-00792-z). This version employs a multiscale deep generative model to predict ligand poses within an existing pocket of the input protein structure. The model outputs new protein conformations along with ligand poses. Therefore, it is categorized as a DL docking method rather than a co-folding model.
>
> ---
>
> **Q:** Multi-ligand complexes. How were cases with cofactors/ions/secondary ligands handled during inference and evaluation—were additional ligands kept, and if not, how might their absence affect PB-valid?
>
> **A:** The PoseBusters dataset includes small molecules within 5Å of the ligand of interest, whereas other datasets do not. Thus, for PoseBusters, the highest-quality dataset, the overlap filter with surrounding molecules functions as intended, while for other datasets it does not. Since overlap with surrounding molecules is partially correlated with RMSD, predictions with high RMSD generally fail both the RMSD and overlap filters. Therefore, the absence of small molecules in those datasets is unlikely to critically affect PB-valid outcomes.
>
> ---
>
> **Q:** Boltz-2 PB-validity. PB-validity appears low for Boltz-2 in several subsets. This is directly opposite result compared to the Boltz-1x utilizing FK-steering with PB potential. Can the authors provide more details about running Boltz-2?
>
> **A:** We thank the reviewer for pointing out these parameter-specific details. We reran Boltz-2 with pocket conditioning enabled, which substantially improved its performance. We have updated the text and corrected our conclusions regarding this tool.
>
> ---
>
> **Q:** The text states Gnina "was not trained on PDBBind or Binding MOAD.", but they actually used their own CrossDocked dataset, which curated from PDBBind dataset. Please clarify the exact training dataset and provide a citation.
>
> **A:** The training set of the recent Gnina version differs from those used by other tools (PDBBind and BindingMOAD). Gnina was (re)trained on the CrossDocked2020 dataset, which was assembled from PDB structures and uses affinity annotations from PDBBind. While we accounted for similarity to the standard training sets, we did not evaluate similarity to CrossDocked2020. We expect limited impact on our analyses, as approximately 40% of PDBBind proteins map to CrossDocked2020. We have now added the correct training dataset information for Gnina and provided the appropriate citation.
>
> ---
>
> **Q:** Why Matcha excels on highly flexible ligands? Can the authors provide an ablation/hypothesis (architecture, sampling, scoring) explaining Matcha's favorable behavior on flexible ligands?
>
> **A:** Thank you for raising this important point. Key hypothesis: Matcha’s generative sampling complexity scales approximately linearly with the number of torsional degrees of freedom, whereas search-based methods such as smina/Vina exhibit exponential complexity growth. This difference likely explains Matcha’s stronger performance on highly flexible ligands.
>
> ---
>
> **Q:** Details about experiment for different ligand classes. How did authors predict the cofactor ligands? PoseBusters dataset contains many multi-ligand systems with cofactor and small molecule ligand. In this system, did authors only used cofactor, which consists of a single heavy atom, rather than using all non-polymer entities in the system?
>
> **A:** Cofactors were manually assembled based on the list of known cofactors from DrugBank and PDB annotations (the code is provided in the submission). In PoseBusters, cofactors appear alongside the ligand of interest and are used to compute the overlap between predicted conformations and nearby molecules. For ligand classification, we used the ligand of interest to determine its membership in each ligand class.
>
> ---
>
> **Q:** Odd inequality symbols in the ligand-similarity thresholds in Appendix B.3.
>
> **A:** Thank you for catching this typographical error. It has been fixed.

---

### Official Review · Reviewer_55Xs · 2025-10-31

**Soundness:** 3
**Presentation:** 2
**Contribution:** 2
**Rating:** 4
**Confidence:** 3

**Summary:**

This paper presents BENTO, a comprehensive benchmark that evaluates 11 protein-ligand interaction prediction tools, spanning classical physics-based methods, various deep learning (DL) models, and co-folding approaches. The work does not introduce new raw data but instead curates and re-partitions existing, well-known datasets (PDBBind Timesplit, PoseBusters, Astex, DockGen). The core contribution is the systematic analysis of tool performance across multiple subsets stratified by ligand class, ligand complexity, drug-likeness, and, crucially, the similarity of both protein pockets and ligands to common training sets. The study aims to disentangle these factors to provide a more nuanced understanding of tool performance, particularly concerning generalization to novel targets and applicability in real-world drug design scenarios.

**Strengths:**

1. The benchmark is extensive, evaluating a wide and timely range of 11 different tools, including very recent methods like AlphaFold3, FlowDock, and Matcha, which provides a valuable, up-to-date comparison.
2. The primary strength lies in the meticulous curation of datasets into focused subsets. This stratification allows the authors to move beyond simplistic overall performance metrics and provide deeper insights into how factors like ligand complexity or pocket similarity specifically impact the performance of different classes of tools.
3. The paper provides practical takeaways for the community. For example, it highlights AlphaFold3's strength on complex molecules, Gnina's impressive robustness and performance on drug-like tasks, and the persistent generalization challenges for many pure DL-based methods. These findings are useful for guiding researchers in selecting the appropriate tool for their specific task.

**Weaknesses:**

1.  The benchmark is constructed entirely from previously published datasets. While the curation and analysis are valuable, the work does not contribute new primary data to the field. Furthermore, some of the conclusions, while systematically demonstrated here across a broader set of tools, reinforce trends already known in the community. For instance, the observation that DL models can struggle to generalize to proteins with binding pockets dissimilar to their training set has been a key takeaway from prior benchmarks like DockGen and PoseBusters. Similarly, the finding that larger and more flexible ligands are inherently more difficult to dock is a well-established principle.
2.  The paper mentions other benchmarks in the introduction, but it lacks a formal "Related Work" section. Such a section would be crucial for formally positioning BENTO within the existing landscape of benchmarking efforts. It would allow the authors to more clearly contrast their methodology and objectives with preceding work and better articulate the specific gaps their benchmark aims to fill.
3.  The visual presentation of results needs improvement. In several key figures (e.g., Figures 2, 4, 5, and others), the text labels, numbers, and data points are overlapping and crowded. This makes the plots difficult to read and interpret, detracting from the paper's ability to clearly communicate its quantitative findings. Given that a benchmark paper relies heavily on graphical comparisons, ensuring the clarity of all figures is essential.

**Questions:**

See the weaknesses detailed above. My main suggestions are to more explicitly frame the novelty of the findings against previous work (perhaps in a dedicated Related Work section) and to remake the figures to ensure all text and data points are clearly legible.

---

> ### Author Response · Authors · 2025-12-02
>
> We appreciate the reviewer's suggestions. Below we address them point-by-point:
>
> **Q**: The benchmark is constructed entirely from previously published datasets. While the curation and analysis are valuable, the work does not contribute new primary data to the field. Furthermore, some of the conclusions, while systematically demonstrated here across a broader set of tools, reinforce trends already known in the community. For instance, the observation that DL models can struggle to generalize to proteins with binding pockets dissimilar to their training set has been a key takeaway from prior benchmarks like DockGen and PoseBusters. Similarly, the finding that larger and more flexible ligands are inherently more difficult to dock is a well-established principle.
>
> **A**: We thank the reviewer for pointing out that we did not clearly highlight the novelty of our work. We have reformulated our key findings and emphasized how our study differs from prior benchmarks.
> We demonstrate the importance of careful dataset curation and stratification across multiple factors when working with multifactor data such as protein-ligand complexes. Specifically, when constructing a dataset to evaluate generalization to unseen pockets, proteins should be filtered explicitly based on pocket structural similarity rather than by PDB release date. Additionally, ligand complexity must be controlled independently, as it can otherwise confound assessments of pocket-level generalization. Our expanded analysis shows that both classical and DL-based docking tools perform well on drug-design-relevant datasets. However, under pocket-aware evaluation, all tools (physics-based, co-folding, and DL-based) perform similarly poorly on protein pockets that are structurally dissimilar to those in their training data. For structurally complex ligands, co-folding tools generally outperform other approaches, whereas most tools achieve comparable accuracy on typical small-molecule ligands.
> We also added a Related Work section, where we outline the limitations of prior benchmarks and explain how our study addresses these gaps.
>
> ---
>
> **Q**: The paper mentions other benchmarks in the introduction, but it lacks a formal "Related Work" section. Such a section would be crucial for formally positioning BENTO within the existing landscape of benchmarking efforts. It would allow the authors to more clearly contrast their methodology and objectives with preceding work and better articulate the specific gaps their benchmark aims to fill.
>
> **A**: We have added a "Related Work" section that summarizes the key findings and limitations of previous benchmarks, and we have highlighted the strengths and unique contributions of our study compared to earlier works.
>
> ---
>
> **Q**: The visual presentation of results needs improvement. In several key figures (e.g., Figures 2, 4, 5, and others), the text labels, numbers, and data points are overlapping and crowded. This makes the plots difficult to read and interpret, detracting from the paper's ability to clearly communicate its quantitative findings. Given that a benchmark paper relies heavily on graphical comparisons, ensuring the clarity of all figures is essential.
> **A**: We have reorganized the figures by grouping tools according to statistically significant differences and adjusted the labels to reduce crowding. The updated plots are now more readable and clearly present the quantitative results.

---

### Official Review · Reviewer_jdjK · 2025-11-01

**Soundness:** 3
**Presentation:** 4
**Contribution:** 3
**Rating:** 6
**Confidence:** 5

**Summary:**

This paper introduces BENTO, a comprehensive benchmark that evaluates 11 diverse docking and co-folding methods for protein–ligand interaction prediction. The benchmark combines classical physics-based tools, deep learning-based docking methods (Gnina, DiffDock, Matcha), and co-folding approaches. Unlike prior benchmarks, BENTO emphasizes curated subsets stratified by ligand physicochemical properties, binding pocket similarity, and protein–ligand pair diversity.

The results are clearly analyzed: physics-augmented and hybrid methods perform most robustly on unseen pockets, while deep learning methods tend to overfit. The analysis also identifies distinct behavior between protein-similar and pocket-dissimilar cases, providing valuable insight into model generalization.

Overall, this paper is a valuable addition to the docking-benchmarking landscape, though some conceptual gaps and missing discussions limit its completeness.

**Strengths:**

**Originality:** The integration of pocket- and ligand-based stratification into a unified benchmark is novel. The combination of classical, deep learning, and co-folding methods in a single framework is also valuable.

**Quality:** The methodological design is thorough — dataset preprocessing, G-LoSA pocket analysis, and ligand categorization (rotatable bonds, molecular size, etc.) are well-documented.

**Clarity:** The paper is written clearly and logically. Figures and tables are informative, particularly the comparative results.

**Significance:** The study contributes to the evaluation of generalization in docking models and provides an extensible foundation for future benchmarks. Its insights are useful for both ML researchers and computational chemists developing next-generation docking systems.

**Weaknesses:**

**Originality:** While comprehensive, the benchmark overlaps heavily with prior datasets already used in [1,2,3]. The novelty mainly lies in combining existing testbeds under one evaluation protocol. No new metric or dataset is introduced.

**Quality:** The main methodological limitation is the single-pocket assumption. All evaluations assume a known binding site, whereas realistic docking and screening pipeline should consider multiple potential pockets[3].

A second issue concerns partial dataset coverage, particularly the Timesplit test set (reported as “332/363” docked complexes).
It remains unclear whether the exclusion of 31 complexes was due to dataset preprocessing (e.g., missing coordinates, cofactors, or large ligands) or tool-specific failures.

If the filtering was not uniform across all methods, this may unintentionally bias the benchmark toward easier systems and underestimate performance variability across tools.

Clarifying the criteria for these exclusions and ensuring consistent evaluation coverage would enhance the fairness and transparency of the benchmark.

Additionally, the inclusion of Matcha (an unreleased method under concurrent review) raises questions of reproducibility and fairness. It is unclear whether the model weights or code were publicly available at the time of evaluation.

**Clarity:** Some terms are used too broadly (e.g., “co-folding” to describe distinct architectures like AlphaFold3 and Boltz-2). Figures are occasionally dense and could benefit from simplified legends.

**Significance:** While valuable as a standardized benchmark, the study’s impact is incremental. It reinforces known findings — physics-based methods remain robust [3]; DL models overfit — without deeply analyzing why certain architectures generalize better.

**Suggestions:**

I think the authors can improve the paper by:

- Extending the benchmark to multi-pocket or blind docking scenarios, which are closer to real-world drug discovery.
- Providing rationale for coefficient choices in RMSD/PB-valid scoring and thresholds.
- Simplifying figure layouts by summarizing core results in one overview chart.
- Including future directions on integrating pocket ensembles or multiple binding sites.

**References:**

[1] Cao, D., Chen, M., Zhang, R. et al. SurfDock is a surface-informed diffusion generative model for reliable and accurate protein–ligand complex prediction. Nat Methods 22, 310–322 (2025). https://doi.org/10.1038/s41592-024-02516-y

[2] Jiang Z. et al., PoseX: AI Defeats Physics Approaches on Protein-Ligand Cross Docking. arXiv 2025, arXiv:2505.01700

[3] Sarigun A. et al., PocketVina Enables Scalable and Highly Accurate Physically Valid Docking through Multi-Pocket Conditioning. arXiv 2025, arXiv:2506.20043

**Questions:**

- How was Matcha evaluated given that it is “under review” at ICLR 2026? Were pre-release weights provided by the authors or independently reproduced?

- Could the authors discuss how results might change in multi-pocket or blind docking settings where the binding site is not known a priori?

- Are the observed performance differences statistically significant (e.g., using paired bootstrap or t-tests)?

- Given the overlap with previous benchmarks, what does BENTO uniquely reveal about generalization that prior work (e.g., PocketVina or DockGen) did not?

- Would the authors consider releasing per-target breakdowns or including docking time/performance trade-offs in supplementary material?

**Details Of Ethics Concerns:**

The paper includes benchmarking of Matcha, a method that is currently also under review for ICLR 2026 and not yet publicly released.
According to the text (Section 2.1), “an anonymized version of the manuscript describing Matcha is provided in the supplementary material.”

This suggests that the authors had privileged access to an unreleased concurrent submission, which may raise fairness and reproducibility concerns.

While this does not indicate any misconduct or data misuse, it could disadvantage other teams who do not have access to that method.

I am flagging this as a responsible research practice / fairness issue rather than a serious ethics violation.

---

> ### Author Response · Authors · 2025-12-02
>
> We appreciate the reviewer's feedback on generalization, which helped us clarify the relevant sections. As suggested, we have added discussion of multi-pocket and blind docking scenarios, provided rationale for coefficient choices in RMSD/PB-valid scoring and thresholds, and simplified figure layouts. As for partial dataset coverage, it comes from tests processing procedures (removal of duplicated entries across datasets, complexes with peptides longer than eight amino acids, inorganic ligands, and entries that are not protein-ligand complexes). We describe it in detail in Appendix A.
>
> Below are our point-by-point responses:
>
> ---
>
> **Q**: How was Matcha evaluated given that it is “under review” at ICLR 2026? Were pre-release weights provided by the authors or independently reproduced?
>
> **A**: The weights and inference code were provided directly by the authors.
>
> ---
>
> **Q**: Could the authors discuss how results might change in multi-pocket or blind docking settings where the binding site is not known a priori?
>
> **A**: Multi-pocket docking is indeed a relevant consideration, but neither our benchmark nor previous studies have addressed it systematically. We anticipate that available data would be insufficient and highly imbalanced, likely overrepresenting certain pockets and forcing the tools to prioritize one pocket over another. For blind docking, our experiments with physics-based tools and Boltz-2 show a considerable drop in performance when the pocket is unspecified, highlighting the limited pocket-search capabilities of most docking tools. Tools operating in blind mode generally underperform compared to those using pocket information. The only exception is AlphaFold3, which operates exclusively in blind-pocket mode and demonstrates strong overall performance, though it still struggles on previously unseen pockets.
>
> ---
>
> **Q**: Are the observed performance differences statistically significant (e.g., using paired bootstrap or t-tests)?
>
> **A**: We thank the reviewer for this important question. We conducted paired t-tests and updated our conclusions based on statistically significant differences between tool performances. Additionally, we grouped bars in figures according to statistical significance, improving readability.
>
> ---
>
> **Q**: Given the overlap with previous benchmarks, what does BENTO uniquely reveal about generalization that prior work (e.g., PocketVina or DockGen) did not?
>
> **A**: Prior benchmarks addressing generalization to new pockets often did not explicitly control for structural similarity between test and training pockets (e.g., PoseBusters used a release-date cutoff) or did not account for ligand complexity, confounding protein-level results (e.g., DockGen).
>
> In contrast, BENTO 1) explicitly controls for pocket similarity using G-Losa, and 2) excludes ligand-level effects.
>
> Our analyses show that, in a pocket-aware scenario, physics-based, co-folding, and DL tools all fail similarly on dissimilar pockets. This finding aligns with PoseX results, albeit using a different metric (average RMSD) and a different similarity-control procedure (TM-score).
>
> ---
>
> **Q**: Would the authors consider releasing per-target breakdowns or including docking time/performance trade-offs in supplementary material?
>
> **A**: We have added corresponding data to the attachment to submission and add discussion of trade-offs to the manuscript.

---

### Author Response · Authors · 2025-12-02

We sincerely thank all reviewers for their careful evaluation and insightful feedback. Your comments have been invaluable in improving the quality, clarity, and rigor of our manuscript. Below, we summarize the key revisions and enhancements made in response to your suggestions.

**Key Improvements in the Manuscript:**

1. **Emphasized the novelty of our approach and key findings.**  As noted by reviewers i6gM, 55Xs, and jdjK, our initial manuscript did not sufficiently highlight the originality of our contribution. We have now reformulated and clarified our core findings:
We demonstrate the importance of careful dataset curation and stratification across multiple factors when working with multifactor data such as protein-ligand complexes. Specifically, when constructing a dataset to evaluate generalization to unseen pockets, proteins should be filtered explicitly based on pocket structural similarity rather than by PDB release date. Additionally, ligand complexity must be controlled independently, as it can otherwise confound assessments of pocket-level generalization. Our expanded analysis shows that both classical and DL-based docking tools perform well on drug-design-relevant datasets. However, under pocket-aware evaluation, all tools (physics-based, co-folding, and DL-based) perform similarly poorly on protein pockets that are structurally dissimilar to those in their training data. For structurally complex ligands, co-folding tools generally outperform other approaches, whereas most tools achieve comparable accuracy on typical small-molecule ligands.
We also added a Related Work section, where we outline the limitations of prior benchmarks and explain how our study addresses these gaps.

2. **Computed statistical significance between tool predictions.** Following the suggestion of reviewer jdjK, we computed statistical significance across tools’ predictions using paired t-tests. We revised our conclusions where appropriate. Details of the procedure are provided in Appendix B.4, and we grouped tools in figures according to significant differences to improve interpretability.

3. **Improved figure readability.** As noted by reviewers 55Xs and jdjK, several plots were overcrowded. We reorganized the figures by grouping tools based on statistical significance and adjusting labels, resulting in clearer and more readable visualizations.

4. **Added predictions for pocket-conditioned Boltz-2.** Reviewer jdjK pointed out that Boltz-2 supports pocket-conditioned inference. We reran Boltz-2 with pocket information supplied, which led to a substantial improvement in its performance. The updated results and revised conclusions are now reflected in the manuscript.

We believe that these revisions and additions substantially enhance the quality of the paper. We sincerely thank you again for your time and thoughtful feedback, which have been invaluable in improving this work.

---

### Meta-Review · Area_Chair_RDPU · 2026-01-08

**Summary:**

1. lack of novelty. (*jdjK*,*55Xs*,*i6gM*)  The findings and analysis do contribute new understanding to the space. (*jdjK*, *55Xs*)
2. issues with dataset coverage and consistency including single-pocket assumption, partial dataset coverage, non-uniform filtering (*jdjK*).
3. clarity of writing and figures (*jdjK*, *55Xs*) and missing experimental details (*i6gM*)
4. there is no related work section (*55Xs*)
5. bias introduced by P2Rank. Should test with and without pocket conditioning. (*i6gM*)

**Reviewer Concerns:**

1. The authors point to evaluating generalization in terms of pocket structural similarity and ligand complexity. Their insights involve superior performance of all tools on drug-design tasks and poorly on pocket-aware evaluation.
2. The authors point out multi-pocket data is not available in adequate quantity and quality.
3. The authors redid the figures to improve clarity and added additional experimental details to the manuscript.
4. The authors added a related work section.
5. The authors added new results as requested.

**Reviewer Scores:**

- *jdjK* gave a 6 and probably would have kept it.
- *55Xs* gave a 4.  There is a small chance they may have increased their score.
- *i6gM* gave a 4.  There is a small chance they may have increased their score.
- *xgwq* gave a 6 and probably would have kept it.

---

### Decision · Program_Chairs · 2026-01-26

Reject